# HIV-1 integrase resistance associated mutations and the use of dolutegravir in Sub-Saharan Africa: A systematic review and meta-analysis

**Ezechiel Ngoufack Jagni Semengue**[1,2,3]*, **Maria Mercedes Santoro**[3], **Valantine Ngum Ndze**[4☯], **Aude Christelle Ka'e**[1,2,5☯], **Bouba Yagai** [1], **Alex Durand Nka**[1,2,3], **Beatrice Dambaya**[1], **Desiré Takou**[1], **Georges Teto**[1], **Lavinia Fabeni**[6], **Vittorio Colizzi**[1,2,3,7], **Carlo-Federico Perno**[1,8], **Francesca Ceccherini-Silberstein**[3,5], **Joseph Fokam** [1,4,9,10]*

**1** Chantal Biya International Reference Center for Research on HIV/AIDS Prevention and Management, Yaoundé, Cameroon, **2** Evangelical University of Cameroon, Bandjoun, Cameroon, **3** University of Rome "Tor Vergata", Rome, Italy, **4** Faculty of Health Sciences, University of Buea, Buea, Cameroon, **5** Doctoral School of Microbiology, Immunology, Infectious Diseases and Transplants, MIMIT, University of Rome "Tor Vergata", Rome, Italy, **6** Laboratory of Virology, National Institute for Infectious Diseases "Lazzaro Spallanzani" -IRCCS, Rome, Italy, **7** Chair of Biotechnology-UNESCO, University of Rome "Tor Vergata", Rome, Italy, **8** Bambino Gesu Children's Hospital, Rome, Italy, **9** Faculty of Medicine and Biomedical Sciences, University of Yaounde I, Yaounde, Cameroon, **10** National HIV Drug Resistance Working Group, Ministry of Public Health, Yaounde, Cameroon

☯ These authors contributed equally to this work.
* josephfokam@gmail.com (JF); ezechiel.semengue@gmail.com (ENJS)

ⓐ OPEN ACCESS

## Abstract

As sub-Saharan Africa (SSA) countries are transitioning to dolutegravir (DTG)-based ART, baseline data are required for optimal monitoring of therapeutic response. In this frame, we sought to generate up-to-date evidence on the use of integrase-strand transfer inhibitors (INSTI) and associated drug resistance mutations (DRMs) within SSA. In this systematic review and meta-analysis, we included randomized and non-randomized trials, cohort-studies, cross-sectional studies, and case-reports published on INSTI or integrase DRMs in SSA. We included studies of patients exposed to DTG, raltegravir (RAL) or elvitegravir (EVG). Primary outcomes were "the rate of virological control (VC:<50copies/ml)" and "the presence of DRMs" on INSTI-based regimens among patients in SSA. We synthesised extracted data using subgroup analysis, and random effect models were used where appropriate. Additional analyses were conducted to assess study heterogeneity. We identified 1,916 articles/citations through database searches, of which 26 were included in the analysis pertaining to 5,444 patients (mean age: 37±13 years), with 67.62% (3681/5444) female. Specifically, 46.15% (12/26) studies focused on DTG, 26.92% (7/26) on RAL, 23.08% (6/26) on both DTG and RAL, and 3.85% (1/26) on EVG. We found an increasing use of DTG overtime (0% before 2018 to 100% in 2021). Median treatment duration under INSTI-based regimens was 12 [9–36] months. Overall, the rate of VC was 88.51% [95%CI: 73.83–97.80] with DTG vs. 82.49% [95%CI: 55.76–99.45] and 96.55% [95%CI: 85.7–100.00] with RAL and EVG, respectively. In univariate analysis, VC with DTG-containing vs. other INSTI-regimens was significantly higher (OR = 1.44 [95%CI: 1.15–1.79], p = 0.0014). Among reported

**Data Availability Statement:** All studies included in this review are listed in Table 1 within the paper. Studies excluded are listed in the supporting document (S1 Appendix) attached to this paper.

**Funding:** The study was financially supported by the CIRCB's annual budget plan 2020-2022. The funders had no role in study design, data collection and analysis, decision to publish, or preparation of the manuscript.

**Competing interests:** The authors have declared that no competing interests exist.

DRMs at failure, the only DTG resistance-mutations were G118R and R263K. In SSA, DTG presents a superiority effect in VC compared to other INSTIs. Nonetheless, the early detection of INSTI-DRMs calls for sentinel surveillance for a successful transition and a sustained efficacy of DTG in SSA.

**PROSPERO Registration Number**: CRD42019122424.

## Introduction

The current WHO recommendations suggest dolutegravir (DTG)-based antiretroviral therapy (ART) as the preferred first-line regimen to treat HIV in resource-limited settings (RLS) [1–3]. DTG is a second generation integrase strand transfer inhibitor (INSTI), known to be well tolerated, with limited drug interactions, high potency and genetic barrier to resistance, as well as limited risk of cross-resistance with first generation INSTIs (raltegravir–RAL and elvitegravir–EVG) [3–10]. Since its approval in 2013 by the U.S. Food and Drug Administration, DTG's efficacy has been studied in several trials and its superiority proven over other antiretrovirals (ARV) [5, 9, 11], including other INSTIs [3, 7].

An optimised use of DTG in several RLS would contribute substantially in achieving the elimination of HIV by 2030 [12], as per the UNAIDS target of 95-95-95 [13]. Sub-Saharan Africa (SSA) in particular carries the greatest burden of HIV infection (~70% of the global epidemics) with increasing burden of drug resistance (about 1/3 of patients). Also, SSA has a very broad HIV genetic diversity (including both HIV type 1 and type 2), which includes a predominance of HIV-1 non-subtype B viruses with potential effects on ART response and on the patterns of HIV drug resistance (HIVDR)-associated substitutions [4, 14, 15].

Of note, more than 40 substitutions have been associated with the development of resistance to INSTIs in HIV-1 B subtypes [4, 14, 15]. The most prevalent mutations for HIV-1 integrase are at positions 66, 92, 143,147, 148, and 155 [5, 16]. Of note, G118R is a rare mutation in subtype B viruses, which could have an alternative pathway for DTG-resistance selection in non-subtype B viruses [4, 14, 15]. In contrast, R263K is preferentially selected among subtype B viruses as compared to other viral subtypes [4, 14, 15]. As viral subtypes (especially B and C) are known to have differential mechanisms in the selection of drug resistance mutations (DRMs) [5], natural occurring polymorphisms may influence the development of resistance against INSTIs [4, 14, 15].

With as goal to set-up baseline data for an optimal management of people living with HIV in SSA, this systematic review aimed at synthesising evidence on the effectiveness of INSTI as well as integrase resistance-associated mutations commonly found among patients under DTG (or other INSTI)-containing regimens in SSA.

## Materials and methods

The protocol for this systematic review had been registered within PROSPERO under the registration number N˚CRD42019122424 and published online [17]. This systematic review is reported in accordance with the reporting guidance provided in the Preferred Reporting Items for Systematic Reviews and Meta-Analyses (PRISMA) statement (see S1 Checklist).

### Study design and eligibility criteria

For this systematic review and meta-regression analysis, we included randomized and non-randomized trials, cohorts, cross-sectional studies, and case reports underlining integrase

resistance-associated mutations or evaluating treatment outcomes under INSTI-containing regimens among HIV-infected patients in SSA. Eligible studies involved INSTI-experienced participants of both gender living in SSA. Studies focusing on patients under DTG-containing regimens were considered as our main group of interest and those focusing on RAL- or EVG-containing regimens served as comparators. We conducted the search on PubMed, Cochrane Central Register of Controlled Trials (CENTRAL), Latin American and Caribbean Health Sciences Literature (LILAC), Web of Science, African Journals Online and Cumulative Index to Nursing and Allied Health Literature (CINAHL) databases. We supplemented the systematic review with hand searching of reference lists of relevant reviews and trials in grey literature and governmental proceedings and also looked for conference abstracts. The search was restricted to articles published in English and French, since the approval of RAL to treat HIV infection in 2007 to May 2021. We excluded studies of patients with specific co-morbidities such as tuberculosis or opportunistic infections. Complete search strategy is described in our systematic review and meta-analysis protocol [17].

The following data were extracted from each study: first author, title of the paper, year of publication, country, sex, INSTI-based regimens, treatment duration, baseline and final viremia, the rate of viral suppression (viremia <1.000 copies/ml), CD4 cells before and after INSTI (wherever reported), the proportion of patients with immune recovery, proportion of adverse reactions (wherever reported), HIV subtype, integrase-DRMs and polymorphisms. Importantly, alongside viral suppression, several studies we included reported patients achieving virological control (VC; viremia<50 copies/ml). Additionally, the effect of each INSTI (DTG, RAL and EVG) on VC and immune recovery (>500 cells/mm$^3$) was evaluated whenever possible. ENJS and ADN did the searches; ENJS and ACK extracted the data and all disagreements were resolved by consensus through discussion with BY, VNN or JF. The database was manually scanned for duplicates by ENJS, ACK and ADN. Where there were duplicate publications, the publication with more detailed information was used.

## Data analysis and synthesis

The two major aims of our study were to inform on the suitability of DTG-use and to characterize the genotypic profile of patients under INSTI-containing regimens in the frame of ART transition in SSA [1–3]. We therefore evaluated treatment outcomes (principally VC and immune recovery) according to INSTI-regimens, treatment duration, gender, age, countries, and we reported INSTI-DRMs wherever described.

To estimate the heterogeneity between studies, $I^2$ and H statistics were used [18]. The $I^2$ value was indicative of the degree of heterogeneity with values of 0%, 18%, 45%, and 75% designated none, low, moderate and high heterogeneity respectively [19]. Lack of evidence on heterogeneity between studies was indicated by obtaining H values close to 1 and these values were inversely correlated with degree of heterogeneity. Prevalence, 95% confidence intervals (95% CI), and prediction intervals were estimated by random effect models [20]. During subgroup and meta-regression analyses, the country, gender, age, pregnancy (wherever applicable) and treatment duration were employed to adjust the variations in pooled estimates. Dependent variables were the rate of virological control, the rate of immune recovery and the presence of INSTI-DRMs.

Statistical significance threshold was set at 0.05. The publication bias was assessed by visual inspection of the asymmetry of the funnel plot [21] and the Egger test with the value of p <0.1 indicating a potential bias [22]. The R version 3.6.0 software (packages "meta" and "metafor") through the RStudio interface was used to perform all meta-analysis [23, 24]. Finally, we used the Grading of Recommendations Assessment, Development and Evaluation (GRADE) system for rating overall quality of evidence.

## Results

We identified a total of 1,916 articles and citations through database searches and other sources. Of these, 234 were selected for eligibility assessment and/or full-text review (Fig 1). Ultimately 26 articles were included in the analysis [25–50] pertaining to seven randomized studies, three non-randomized studies, five cross-sectional studies, three cohorts and eight case reports. These studies included 5,444 patients aged from 17–70 (mean: 37±13) years (See Table 1). Among these patients, 67.62% (3681/5444) were registered as females and the rate of missing information on gender was 17.96% (978/5444). Overall, trials included in this analysis (both randomized and non-randomized) had low risk of bias, with moderate to high quality of evidences while cross-sectional and cohort studies had moderate quality of evidences and case reports were rated with low quality of evidences (Table 2). Egger's regression test did not show any evidence for publication bias among the included studies (Fig 2).

Regarding the geographical location, most included studies were multi-centric, carried out across different countries at the same time. However, South Africa hosted most of the studies (9), followed by Uganda (6), Botswana (5), Kenya and Zimbabwe (4 each), Cameroon (3), Malawi and Senegal (1 each). All of SSA was therefore represented, but in various proportions and different study designs were being carried out (Fig 3).

Though some studies were exclusively focused on women at different stages of pregnancy [26, 37, 47, 48], none of the studies included in our analysis were restricted to adolescents only. Following INSTI-based regimens, 46.15% (12/26) studies focused on DTG only, 26.92% (7/26) focused on RAL only, 23.08% (6/26) on both DTG and RAL and 3.85% (1/26) on EVG only.

Chronological analysis revealed an increased use of DTG as compared to other INSTI (RAL and EVG) in recent studies (Fig 4). In effect, before 2018, published studies exclusively focused on RAL without any reported case of DTG-exposure. Inversely, the use of DTG increased as from 5/9 (55%) in 2018 to 3/3 (100%) in 2021, with a gradual phase-out of EVG and RAL. Additionally, DTG users were reported either as ART-initiators (see blue bars in Fig 4) or as third-line failures with previous exposure to RAL (see yellow bars in Fig 4).

Baseline viremia and CD4 count were assessed for most, but not all studies included. In effect, the 18/26 studies reporting viremia before INSTI-regimens showed an overall median of 4.28 [3.94–4.70] log copies/ml and the 17/26 studies reporting CD4 count before INSTI-regimens revealed a median of 248 [172–408] cells/mm$^3$. Overall baseline features of study participants are summarized in Table 1.

Based on studies reporting treatment history, the treatment duration under INSTI-containing regimens varied from 2–120 (median: 12 [9–36]) months. After INSTI-regimens, only 12/26 studies reported virological response and most studies reported virological response in terms of VC (<50 copies/ml or <1.69 log copies/ml) rather than viral suppression (<1000 copies/ml or < 3log copies/ml). Among the 12/26 studies reporting virological response, 89.01% [77.87–97.11] reported VC after 2–36 months of exposure to INSTI. Globally, 88.51% [73.83–97.80] of VC was reported in patients after 2–24 months of exposure to DTG; 82.49% [55.76–99.45] after 6–36 months of exposure to RAL and 96.55% [85.79–100.00] after 12 months under EVG (Fig 5). While comparing overall virological response with regards to the exposure to each INSTI in univariate analysis, the use of DTG (7/12 studies) appeared to significantly favor VC (OR = 1.44, 95%CI:[1.15–1.79]; p = 0.0014) while RAL (4/12 studies) seemed to disfavor VC (OR = 0.67, 95%CI:[0.53–0.83]; p = 0.0004). EVG (1/12 study) was also associated with VC but without statistical significance (OR = 4.36, 95%CI:[0.59–32.17]; p = 0.17). No study evaluated the virological response according to age and/or gender distribution. Interestingly, Kenneth Kintu et al., alone reported 74% (89/120) of VC and 93% (112/120) of viral suppression among women at late pregnancy after about two months of exposure to DTG [48].

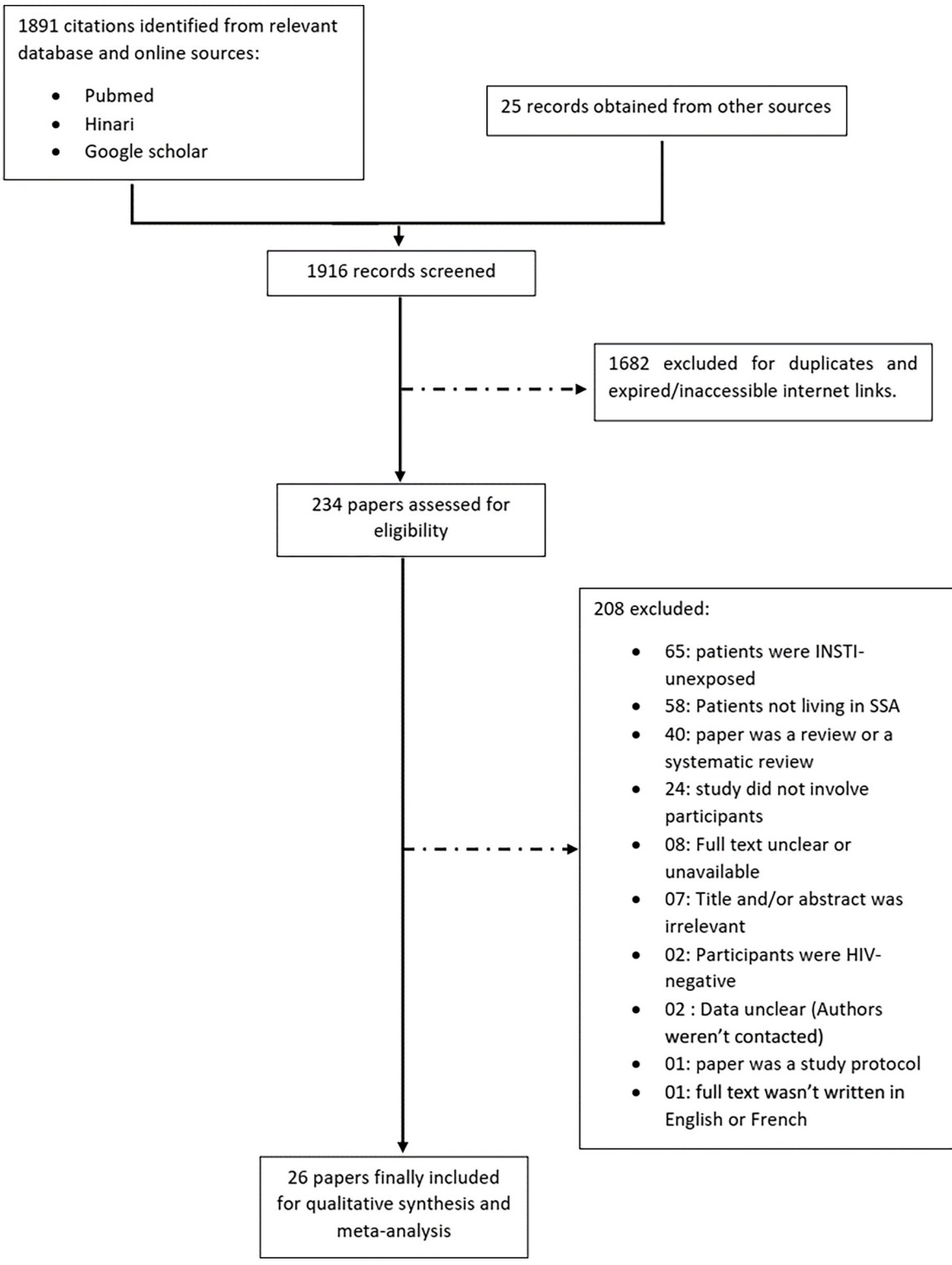

**Fig 1. List of papers excluded from the systematic review and meta-analysis; interrupted lines represent excluded studies with reasons for exclusion.**

**Table 1. Baseline characteristics of study participants.**

| Study reference | Country (ies) | Participants (N) | Mean or Median Age(years) | Male (n) | Female (n) | Study dealing with pregnancy? | Viremia (log copies/ml) | CD4 count (cells/mm³) |
|---|---|---|---|---|---|---|---|---|
| Kaelo K. Seatla, 2021 [25] | Botswana | 34 | 41 | 19 | 15 | No | 4,53 | NR |
| Rebecca Zash, 2018 [26] | Botswana | 1729 | 28 | 0 | 1729 | Yes | NR | 411 |
| Cleophas Chimbetete, 2018 [27] | Zimbabwe | 36 | 41 | 17 | 19 | No | 4,76 | 148 |
| Emmanuel Ndashimye, 2020 [28] | Uganda | 51 | NR | NR | NR | No | NR | NR |
| Selly Ba, 2018 [29] | Senegal | 29 | 49 | 6 | 23 | No | 4,1 | 408 |
| Mahomed Kairoonisha, 2020 [30] | South Africa | 1 | 38 | 0 | 1 | No | 3,45 | 966 |
| Cissy Kityo, 2018 [31] | Kenya, Malawi, Zimbabwe and Uganda | 757 | 36 | NR | NR | No | 5,39 | 37 |
| Venter Willem D.F., 2019 [32] | South Africa | 702 | 32 | 280 | 422 | No | NR | 332 |
| Kouanfack Charles, 2019 [33] | Cameroon | 310 | 37 | 113 | 197 | No | 5,3 | 281 |
| Siedner Mark J., 2020 [34] | South Africa | 557 | 32 | 216 | 341 | No | NR | NR |
| Achieng Loice, 2018 [35] | Kenya | 1 | 70 | 1 | 0 | No | 3,98 | 93 |
| Waitt Catriona, 2019 [36] | Uganda, South Africa | 29 | 27 | 0 | 29 | No | 4,46 | 343 |
| Mmakgomo M. Raesima, 2019 [37] | Botswana | 152 | NR | 0 | 152 | Yes | NR | NR |
| Michelle Moorhouse, 2018 [38] | South Africa | 118 | 41 | NR | NR | No | 4,16 | 172 |
| Theresa M. Rossouw, 2016 [39] | South Africa | 1 | 18 | 0 | 1 | No | 2,69 | 200 |
| Fokam Joseph, 2020 [40] | Cameroon | 1 | 65 | 1 | 0 | No | 5,19 | 108 |
| Ahmed N.S., 2019 [41] | Not properly reported (East of Africa) | 1 | 22 | 0 | 1 | No | 3,93 | NR |
| Emmanuel Ndashimye, 2018 [42] | Uganda | 51 | NR | NR | NR | No | NR | NR |
| Kaelo K. Seatla, 2018 [43] | Botswana | 1 | NR | NR | NR | No | 3,2 | 540 |
| Avelin F. Aghokeng, 2013 [44] | Cameroon | 1 | 45 | 0 | 1 | No | 3,87 | NR |
| Cleophas Chimbetete, 2018 [45] | Zimbabwe | 1 | 17 | 0 | 1 | No | 6 | 200 |
| Kim Steegen, 2019 [46] | South Africa | 43 | 41 | 22 | 21 | No | NR | NR |
| Rebecca Zash, 2018 [47] | Botswana | 426 | NR | 0 | 426 | Yes | NR | NR |
| Kenneth Kintu, 2020 [48] | South Africa and Uganda | 120 | 28 | 0 | 120 | Yes | 4,4 | 464 |
| Nicholas I. Paton, 2021 [49] | Uganda, Kenya, Zimbabwe | 235 | 33 | 95 | 140 | No | 4,5 | 189 |
| Keene, Claire M., 2021 [50] | South Africa | 57 | 37 | 15 | 42 | No | 4,02 | 248 |
| **Overall estimates** | | **5444** | **37** | **785** | **3681** | **/** | **4.28 [3.94–4.70]** | **248 [172–408]** |

**Table 2. Risk of bias assessment.**

| Study reference | Study design | Q1 | Q2 | Q3 | Q4 | Q5 | Q6 | Q7 | Q8 | Q9 | Q10 | Q11 | Q12 | Q13 | Q14 | Q15 | Q16 | Risk of Bias | Quality of evidences |
|---|---|---|---|---|---|---|---|---|---|---|---|---|---|---|---|---|---|---|---|
| Kaelo K. Seatla, 2021 | Cross sectional study (observational/ descriptive study) | | | | | | | | | | | | | | | PY | PN | low | Moderate |
| Rebecca Zash, 2018 | Cross sectional study (observational/ descriptive study) | | | | | | | | | | | | | | | PY | PN | low | Moderate |
| Cleophas Chimbetete, 2018 | Cross sectional study (observational/ descriptive study) | | | | | | | | | | | | | | | PY | PN | low | Moderate |
| Emmanuel Ndashimye, 2020 | Cross sectional study (observational/ descriptive study) | | | | | | | | | | | | | | | PY | PN | low | Moderate |
| Selly Ba, 2018 | Non-randomized study/ trial | | | | | | | No | PN | No | No | Yes | Yes | PN | PY | | | Low | Moderate |
| Mahomed Kairoonisha, 2020 | Case report / Case series | | | | | | | | | | | | | | | No | PY | High | Low |
| Cissy Kityo, 2018 | Randomized study/trial | No | No | Yes | Yes | No | No | | | | | | | | | | | Unclear | Moderate |
| Venter Willem D.F., 2019 | Randomized study/trial | No | PY | PY | Yes | No | PN | | | | | | | | | | | Unclear | Moderate |
| Kouanfack Charles, 2019 | Randomized study/trial | PN | PN | Yes | Yes | No | PN | | | | | | | | | | | Unclear | Moderate |
| Siedner Mark J., 2020 | Randomized study/trial | No | PN | Yes | Yes | PN | No | | | | | | | | | | | Low | High |
| Achieng Loice, 2018 | Correspondance / Letter to editor | | | | | | | | | | | | | | | No | PY | High | Low |
| Waitt Catriona, 2019 | Randomized study/trial | No | PN | PN | Yes | No | No | | | | | | | | | | | Low | High |
| Mmakgomo M. Raesima, 2019 | Cohort study | | | | | | | | | | | | | | | Not clear | PN | Unclear | Moderate |
| Michelle Moorhouse, 2018 | Cohort study | | | | | | | | | | | | | | | Not clear | PN | Unclear | Moderate |
| Theresa M. Rossouw, 2016 | Case report / Case series | | | | | | | | | | | | | | | No | PY | High | Low |
| Fokam Joseph, 2020 | Case report / Case series | | | | | | | | | | | | | | | No | PY | High | Low |
| Ahmed N.S., 2019 | Case report / Case series | | | | | | | | | | | | | | | No | PY | High | Low |
| Emmanuel Ndashimye, 2018 | Cross sectional study (observational/ descriptive study) | | | | | | | | | | | | | | | PY | PN | low | Moderate |
| Kaelo K. Seatla, 2018 | Case report / Case series | | | | | | | | | | | | | | | No | PY | High | Low |
| Avelin F. Aghokeng, 2013 | Case report / Case series | | | | | | | | | | | | | | | No | PY | High | Low |
| Cleophas Chimbetete, 2018 | Case report / Case series | | | | | | | | | | | | | | | No | PY | High | Low |
| Kim Steegen, 2019 | Cohort study | | | | | | | | | | | | | | | Not clear | PN | Unclear | Moderate |
| Rebecca Zash, 2018 | Non-randomized study/ trial | | | | | | | PN | PN | No | Not clear | Yes | Yes | Not clear | No | | | Unclear | Moderate |

*(Continued)*

**Table 2.** (Continued)

| Study reference | Study design | Q1 | Q2 | Q3 | Q4 | Q5 | Q6 | Q7 | Q8 | Q9 | Q10 | Q11 | Q12 | Q13 | Q14 | Q15 | Q16 | Risk of Bias | Quality of evidences |
|---|---|---|---|---|---|---|---|---|---|---|---|---|---|---|---|---|---|---|---|
| Kenneth Kintu, 2020 | Randomized study/trial | No | PN | PN | PY | No | PN | | | | | | | | | | | Low | High |
| Nicholas I. Paton, 2021 | Randomized study/trial | No | No | PY | PN | No | PN | | | | | | | | | | | Low | High |
| Keene, Claire M., 2021 | Non-randomized study/trial | | | | | | | PN | No | No | PN | PN | PN | No | No | | | Low | High |

Q1: According to you is there a risk of bias from the randomization process? Q2: According to you is there a risk of bias in the selection of participants or their adherence to treatment? Q3: Are there missing data concerning primary outcomes? Q4: Are there missing data concerning secondary outcomes? Q5: According to you is there a risk of bias in the method used to measure the outcomes? Q6: Is there a selective reporting of the results according to you? Q7: According to you is there a risk of bias due to confounders in this study? Q8: According to you is there a risk of bias from participant's selection? Q9: According to you is there a risk of bias due to misclassification of intervention in this study? Q10: According to you is there a risk of bias due to treatment adherence in this study? Q11: Are there missing data concerning primary outcomes? (rate of virological control under INSTI-based; level of drug resistance) Q12: Are there missing data concerning secondary outcomes? Q13: According to you is there a risk of bias in the method used to measure the outcomes? Q14: Is there a selective reporting of the results according to you? Q15: According to you does the selection of participants represent general population? Q16: According to you will it be a risk of bias in inferring the results to the general population?

Q1 to Q6 were derived from ROBINS tool and enabled to assess the risk of bias in randomized trials; Q7 to Q14 derived from ROBIS tool and enabled to assess the risk of bias in non-randomized trials; Q15 and Q16 were based on Newcastle-Ottawa scale and enabled to assess the risk of bias in cohorts, cross sectional studies, case controls and case reports. In addition the risk of bias assessment, the GRADE approach was used to give an overall estimate of the quality of the evidences as described in our systematic review protocol [17].

PN: Probably No; PY: Probably Yes.

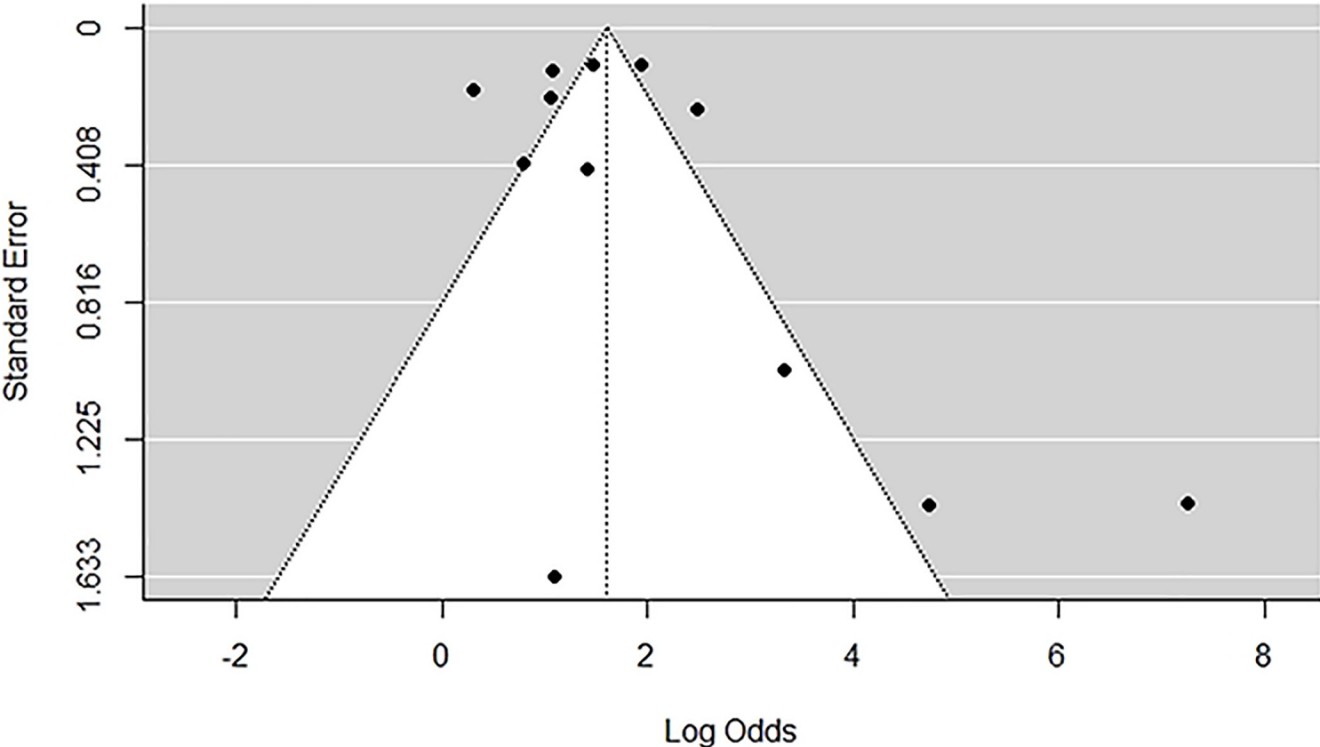

**Fig 2. Funnel plot asymmetry of included studies reporting virological control as primary outcome.** Though on a limited sample size, there was very few outliers.

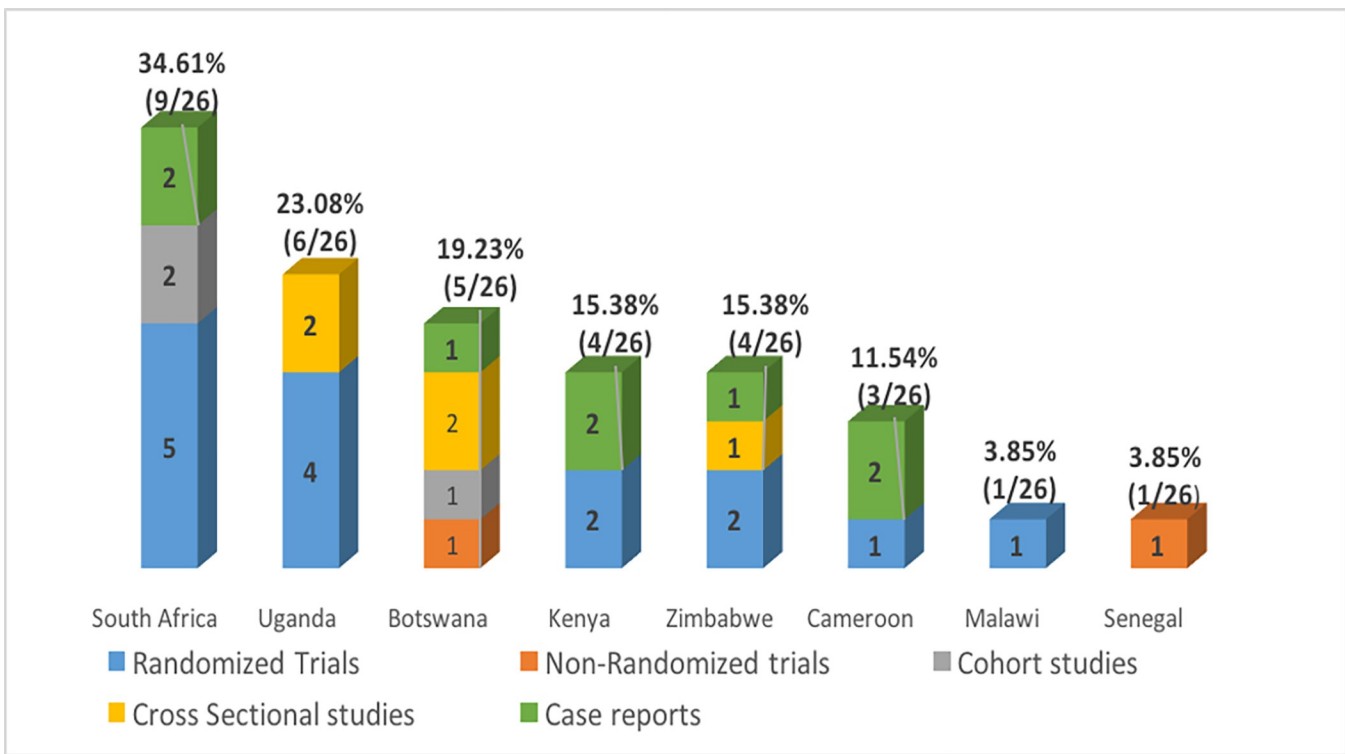

**Fig 3. Distribution of included studies according to countries and study designs; at least one country in every sub-regions of sub-Saharan Africa is represented.**

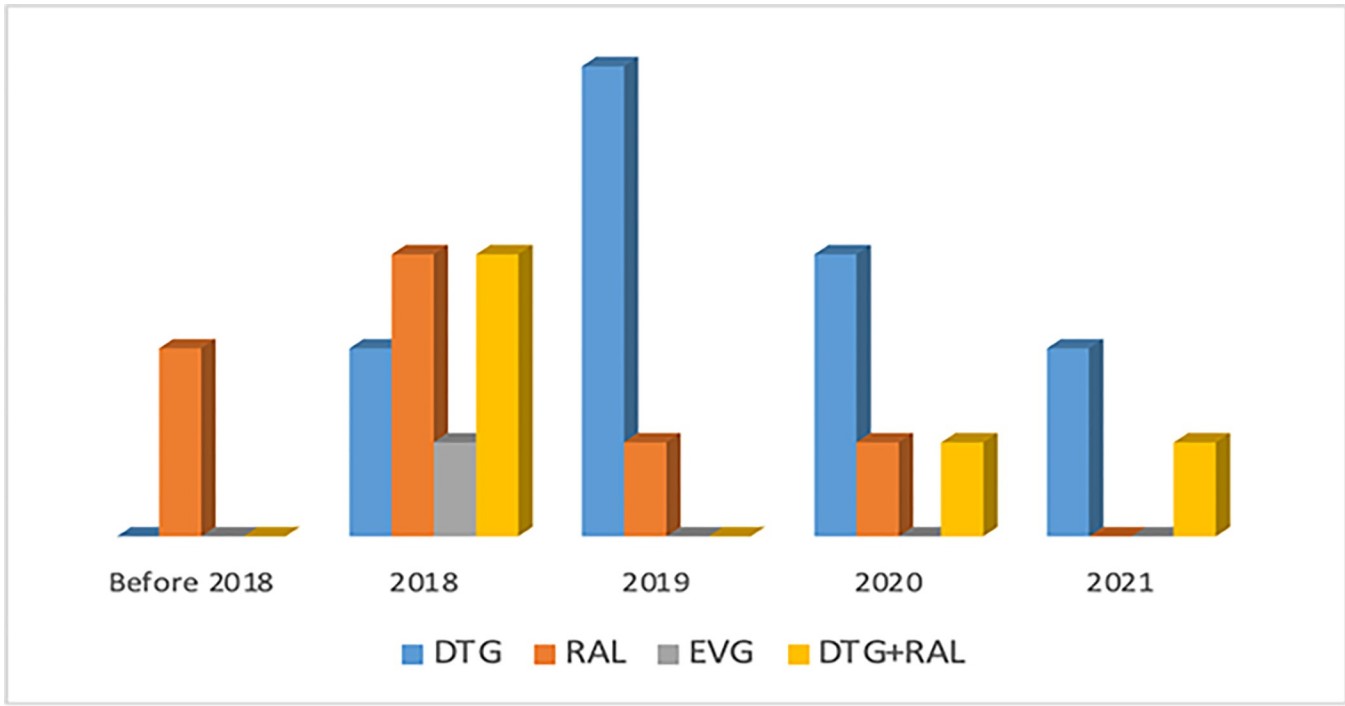

**Fig 4. The figure portrays the use of dolutegravir in sub-Saharan Africa overtime.**

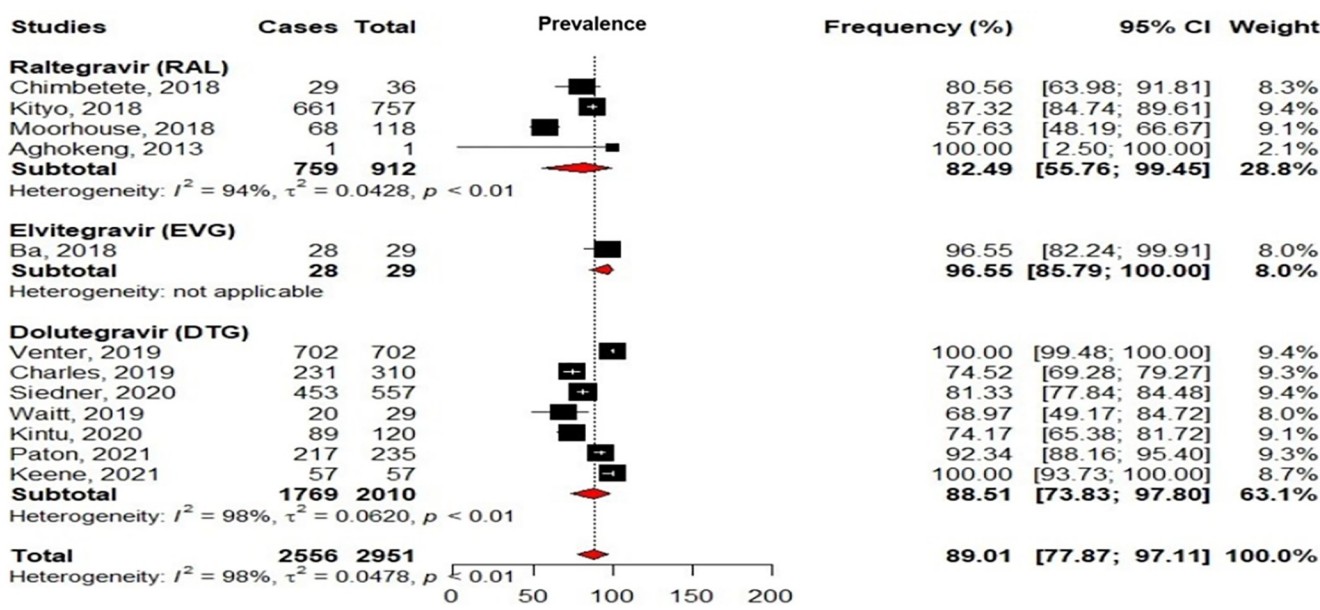

**Fig 5. The plot was constructed with R v3.6.0 software, with the use of "meta" and "metaphor" packages.** Weights are all from random effects analyses and represent the degree of confidence of each study compared to all other studies in the meta-analysis. Squares represent the rate of virological control in each study, with plain lines representing 95%CI in those studies. Diamonds represent pooled estimates within subgroups, with the value being at the center while extremities play the role of 95%CI. The interrupted line represents the median value after aggregation of all the data while the 0–120 scale represents the range of possible data in the analysis.

Studies reporting CD4 cells count on INSTI-regimens were very limited. Of note, the only study reporting immune recovery (from 408 to 569 cells/mm$^3$) was a non-randomized trial carried among HIV-2 infected patients exposed to EVG for 12 months [29]. Two other case reports revealed a slight increase of CD4 cells count in patients infected by HIV-1 subtype C but without prior history of immunological failure in these patients (from 540 to 670 cells/mm$^3$ after subsequent failure to RAL and DTG within a 9 years' timeframe for the first case and from 966 to 1,008 cells/mm$^3$ after 7 months of exposure to DTG for the second) [30, 43]. Four other studies also reported a slight increase of CD4 cells count but without immune recovery among patients exposed to DTG and/or RAL [27, 40, 49, 50]. In the first study (a case report), CD4 increased from 108 to 187 cells/mm$^3$ in a patient failing RAL and DTG after 7 years of exposure to INSTI. The second (a cross-sectional study) reported an increase from 148 to 252 cells/mm$^3$ despite VC after 24 months of RAL-exposure. In the third study (non-randomized trial) CD4 increased from 248 to 373 cells/mm$^3$ and the fourth (randomized trial) reported an increase from 189 to 337 cells/mm$^3$ and both were conducted among patients exposed to DTG for six and 12 months respectively. Finally, two other case reports highlighted a decrease in CD4 cells count among patients consecutively failing RAL and DTG [35, 45]. Putting these data altogether, immune recovery is observed in patients with VC on INSTI-based therapy.

Some studies (6/26) reported adverse events related to DTG-uptake within their settings [26, 30, 32, 37, 47, 48]. The majority of these studies (5/6) involved only female participants at various stages of pregnancy, with summary estimates that were later used for meta-analysis. An overall rate of 10.84% [95%CI: 0.00–41.50] adverse events related to DTG-uptake was found, with essentially mild to severe adverse birth outcomes, insomnia due to posology, and in some instances neural tube defects (Fig 6). Venter Willem D.F. et al., also reported decrease in bone density, reduction of creatinine's clearance and weight disorder among some patients

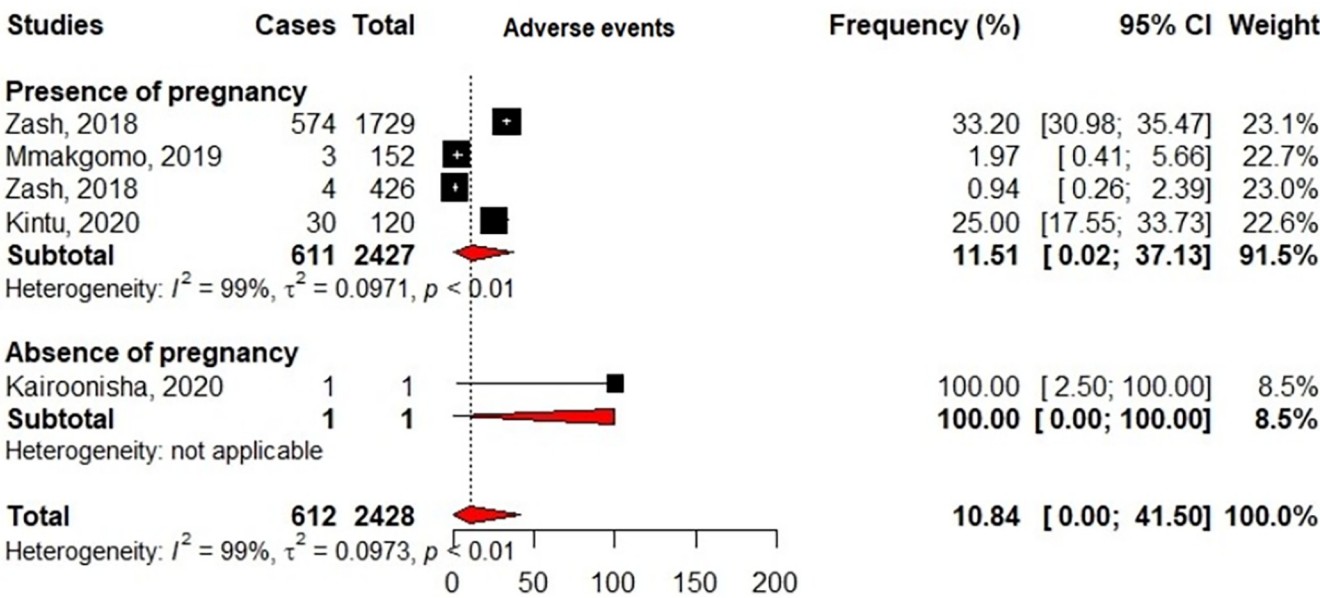

**Fig 6. The plot was constructed with R v3.6.0 software, with the use of "meta" and "metaphor" packages.** Weights are all from random effects analyses and represent the degree of confidence of each study compared to all other studies in the meta-analysis. Squares represent the prevalence of adverse events in each study, with plain lines representing 95%CI in those studies. Diamonds represent pooled estimates within subgroups, with the value being at the center while extremities play the role of 95%CI. The interrupted line represents the median value after aggregation of all the data while the 0–40 scale represents the range of possible data in the analysis.

(both males and females) under DTG-based therapies [32]. All other included studies did not report adverse events under DTG- or INSTI-based within their settings.

Ten studies reported data concerning HIV genetic diversity within their settings [25, 28, 29, 33, 40–44, 46]; we thus confirmed the circulation of HIV-2 in West Africa (Senegal), HV-1 group O, HIV-1 CRF02_AG and CRF18_cpx in Central Africa (Cameroon), mostly HIV-1 subtype C and few HIV-1 subtype G in Southern Africa (South Africa and Botswana), HIV-1 subtype A, subtype C and subtype D in East Africa (Uganda and Kenya).

Regarding INSTI-DRMs, 13 studies highlighted the emergence of both major and secondary mutations, associated to varying levels of INSTI-resistance (Table 3). Reported major INSTI-DRMs included T66A, T66I, T66V, G118R, E138A, E138K, E138Q, G140A, G140S, Y143C, Y143H, Y143R, Y143S, S147G, Q148R, Q148K, N155H, N155D, G163R and R263K whereas secondary DRMs entailed L74I, Q95K, T97A, V151I and E157Q (Fig 7 presents an overview of the distribution across SSA).

## Discussion and conclusions

This systematic review and meta-analysis, aiming at updating the current knowledge on DTG-use in SSA, shows an increasing preferential use of DTG overtime, supported by the implementation of the transition plan to DTG in RLS following the PEPFAR program [1–3, 51]. These findings are in line with previous reports [3–10, 52] and current WHO recommendations [1–3] as they justify the transition to DTG in order to limit the emergence/spread of HIVDR and especially pretreatment NNRTI-DRMs, which varied above the WHO's 10% threshold in many African RLS [53–55]. Secondly, we report the detection of both major and secondary INSTI-DRMs found at a lower burden, thus suggesting little effect on DTG efficacy in SSA at the moment.

Table 3. HIV integrase mutations observed per study and corresponding levels of INSTI-resistance.

| Studies | Reported drug resistance associated mutations | Resistance level[*] | | |
|---|---|---|---|---|
| | | DTG | EVG | RAL |
| Seatla, 2021 [25] | N155H, N155D, Q148R, Q148K, G140A, G118R, T66A, E138K, S147G, E157Q | red | red | red |
| Ndashimye, 2020 [28] | N155H, Q148R, Q148K, G140A, E138A, E138K, Y143R, Y143S | red | red | red |
| Ba, 2018 [29] | Q148R, G140S | orange | red | red |
| Kairoonisha, 2020 [30] | G118R, T66I, E138K | red | red | red |
| Kityo, 2018 [31] | R263K, N155H, Y143R, Y143C, T97A | red | red | red |
| Achieng, 2018 [35] | Q148R, G140A, E138K, S147G, L74I, T97A | red | red | red |
| Rossouw, 2016 [39] | T97A, N155H, V151I, E157Q | green | red | red |
| Fokam, 2020 [40] | Q148R, G140A, E138K, E138Q, S147G, E157Q | red | red | red |
| Ahmed 2019 [41] | R263K | orange | orange | orange |
| Ndashimye, 2018 [42] | N155H, Q148K, Q148R, G140A, T66A, T66I, T66V, E138A, E138K, Y143R, Y143S, S147G, G163R | red | red | red |
| Seatla, 2018 [43] | Q148R, G140A, E138K, S147G | red | red | red |
| Chimbetete, 2018 [45] | Q148R, G140A, E138K | red | red | red |
| Steegen, 2019 [46] | N155H, T66A, E138K, Y143C, Y143R, Y143H, S147G, Q95K | orange | red | red |

[*]Resistance levels to integrase strand transfer inhibitors (INSTI) was interpreted according to the Stanford HIV db algorithm. The "green" refers to the absence or a very low level of resistance; the orange refers to an intermediate level of resistance and the red refers to a high level of INSTI-resistance.

We also reported very high proportions of VC (<50copies/ml) following initiation of INSTI in SSA, with DTG and EVG leading more rapidly to a favorable virological response than RAL. Though these findings are strongly supporting the current trend towards massive adoption of INSTI-based therapies to treat HIV infection (both type 1 and 2), several implications can be drawn depending on HIV types and the investigative INSTI. Selly Ba et al., in their study emphasized on the effectiveness of EVG on HIV-2 within the Senegalese context [29]. They obtained high proportion of VC after 12 months under EVG, even though there are controversial assumptions stipulating that HIV-2 infected patients maintain VC and have undetectable or very low plasma viral loads even in the absence of ART [56]. Interestingly, DTG, RAL and EVG has been further proven to be very potent against HIV-2 infection in previous findings [57–62], supporting the efficacy of all INSTIs in achieving viral control. Additionally, there is immunological recovery in early treatment of HIV-2 patients with low or undetectable plasma viral loads [29]. Similar findings were reported with RAL-containing regimens [27, 31, 38, 44]. Achieving these treatment outcomes requires a reinforced adherence support, close monitoring and sequencing for the surveillance of INSTI-DRMs to ensure long-term efficacy of INSTI-based regimens used in SSA [31, 38, 44]. However, the heterogeneity observed in these studies limited the data aggregation regarding RAL-efficacy. On the other hand, results obtained with the DTG all confirmed its superiority over other INSTIs [32–34, 36, 48–50]. Importantly, burning issues regarding the use of DTG in pregnancy have been adequately addressed and misconceptions have been resolved, with reported events in fetuses not causally correlating to the use of DTG [26, 36, 37, 47, 48]. Two studies demonstrated non-inferiority of DTG to ritonavir boosted darunavir [49] and to low-dose efavirenz [33] and two others focused on the ideal nucleoside reverse transcriptase inhibitors (NRTI) backbone that should be prescribed in combination with DTG [32, 49]. These studies further highlighted the high VC soon after initiation of DTG even with pre-existing NRTI DRMs [34].

Regarding CD4 cells monitoring following exposure to INSTI, studies reporting these data showed immune recovery is a function of baseline CD4 count and a good virological response under an effective INSTI-based therapy in SSA. In effect, immune recovery was observed among patients with mild immunodeficiency at baseline; those severely immune-

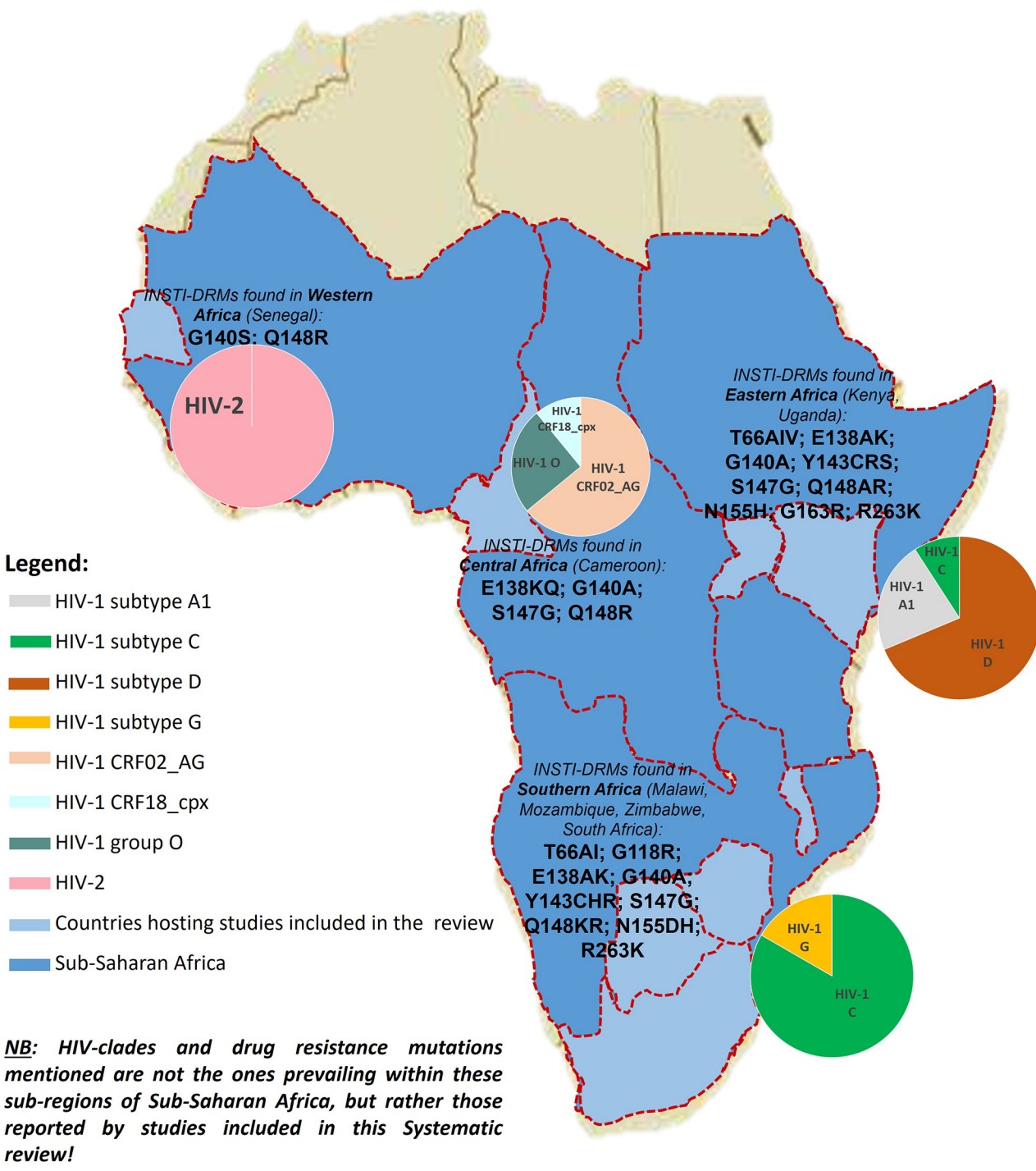

**Fig 7.** (a) The figure was conceived and adapted by the authors of this review; the base layer of the map was downloaded from http://viewer. nationalmap.gov/viewer/. (b) 2022 Semengue et al. This is an open access article published under the Creative Commons Attribution (CC BY) 4.0 license, which means that they will be freely available online, and any third party is permitted to access, download, copy, distribute, and use these materials in any way, even commercially, with proper attribution.

compromised at baseline had a slight improve of their CD4 count under INSTI. Long-term immunological data would however be more informative, especially outcomes among patients being placed on DTG-based following late discovery of HIV infection in SSA [63].

No EVG- or RAL-oriented study reported or focused on adverse events. Reported outcomes were found only in DTG studies, and included mild to severe adverse birth outcomes, insomnia and neural tube defects when taken during pregnancy [26, 36, 37, 47, 48] and decrease in bone density, reduction of creatinine's clearance and weight disorder outside pregnancy [32]. This calls for a thorough pharmacovigilance program while scaling-up access to DTG-containing regimens in SSA.

Remarkably, HIV-clades observed in this systematic review are those reported by studies included in the analysis. Even though we could not specify the prevalence of these clades and thus characterize the burden of HIV-infection in these sub-regions of SSA, our findings correlate with previous reports concerning the high genetic diversity found in these settings. Summary estimates of INSTI-DRMs were limited, underscoring the low-level of DTG major drug resistance within SSA. Nonetheless, some INSTI-DRMs reported are G118R in southern Africa [25, 30] and R263K in East Africa [31, 41]. Of note, if G118R has previously been described as DTG-resistance pathway in non-B (and especially HIV-1 subtype C viruses prevailing in southern Africa [4, 64]), R263K –known as DTG-resistance signature in B-subtype viruses [4, 65–68]–is being reported in East Africa, a region with a prevailing HIV-1 subtype D infection. Cissy Kityo et al. [31], reported the presence of R263K after exposure to RAL, similar to report from Marine Monleau et al. [69], in 2012 (i.e. before approval of DTG by FDA). Interestingly, Cissy Kityo et al. [31], conducted their study in East Africa, were subtype D prevails over HIV-1 subtype C. A concise interpretation of these evidences altogether highlights a possible co-occurrence of R263K and/or G118R, henceforth underscoring a close monitoring and surveillance of INSTI-DRMs among patients on DTG after exposure to RAL for ensuring a successful transition to DTG-based regimens in SSA.

There are some limitations in this study. Firstly, the heterogeneity in data as studies included had different objectives, different targets, and different scopes, thus leading to different end-points for interpretation of data. Secondly, some studies had missing data on primary outcomes and others had small sample size, thus restricting data aggregation. Thirdly, only a single study provided a detailed estimate on side-effect of DTG-based therapy [32], thus limiting the potential to appraise side effect(s) with other INSTIs (RAL and EVG). Fourthly, although the proportion of INSTI-DRMs in SSA is estimated to be low before initiation of INSTI-based therapy [70], a thorough summary estimate of INSTI-DRMs requires further surveillance in SSA. Finally, all studies included in this review did not report data on adherence under INSTI-based therapies. However, considering VC is rapidly achieved, adherence is expected to be very high under DTG; especially as there are fewer drug interactions and tolerability under INSTI-based is known to be beneficial as compared to other drug regimens [71].

Conclusively, our systematic review and meta-analysis revealed increasing use of DTG in recent years in SSA. DTG presents a superiority effect in virological control compared to other INSTIs. However, immune recovery is seemingly comparable across INSTI-regimens. Some mild to severe adverse events have been reported with DTG, highlighting the need of a thorough pharmacovigilance program in SSA. Though at low rates, the early detection of INSTI DRMs calls for sentinel surveillance for a successful transition and a sustained efficacy of DTG across SSA countries.

## Supporting information

**S1 Checklist. PRISMA checklist.**
(PDF)

**S1 Appendix. List of papers excluded in the meta-analysis.**
(DOCX)

## Acknowledgments

We thank the **"Chantal BIYA International Reference Centre"** (CIRCB) for hosting the present study and for all the facilitations.

## Author Contributions

**Conceptualization:** Ezechiel Ngoufack Jagni Semengue, Maria Mercedes Santoro, Valantine Ngum Ndze, Beatrice Dambaya, Carlo-Federico Perno, Francesca Ceccherini-Silberstein, Joseph Fokam.

**Data curation:** Ezechiel Ngoufack Jagni Semengue, Aude Christelle Ka'e, Carlo-Federico Perno, Francesca Ceccherini-Silberstein, Joseph Fokam.

**Formal analysis:** Ezechiel Ngoufack Jagni Semengue.

**Funding acquisition:** Joseph Fokam.

**Methodology:** Ezechiel Ngoufack Jagni Semengue, Valantine Ngum Ndze, Alex Durand Nka, Desiré Takou, Georges Teto.

**Resources:** Ezechiel Ngoufack Jagni Semengue, Alex Durand Nka, Joseph Fokam.

**Software:** Aude Christelle Ka'e.

**Supervision:** Valantine Ngum Ndze, Carlo-Federico Perno, Francesca Ceccherini-Silberstein, Joseph Fokam.

**Validation:** Lavinia Fabeni, Vittorio Colizzi, Carlo-Federico Perno, Francesca Ceccherini-Silberstein, Joseph Fokam.

**Visualization:** Aude Christelle Ka'e, Bouba Yagai, Beatrice Dambaya, Lavinia Fabeni, Vittorio Colizzi, Carlo-Federico Perno, Francesca Ceccherini-Silberstein, Joseph Fokam.

**Writing – original draft:** Ezechiel Ngoufack Jagni Semengue, Maria Mercedes Santoro, Valantine Ngum Ndze, Aude Christelle Ka'e, Bouba Yagai, Alex Durand Nka, Beatrice Dambaya, Desiré Takou, Georges Teto, Lavinia Fabeni, Vittorio Colizzi, Carlo-Federico Perno, Francesca Ceccherini-Silberstein, Joseph Fokam.

**Writing – review & editing:** Ezechiel Ngoufack Jagni Semengue, Maria Mercedes Santoro, Aude Christelle Ka'e, Bouba Yagai, Alex Durand Nka, Beatrice Dambaya, Desiré Takou, Georges Teto, Lavinia Fabeni, Vittorio Colizzi, Carlo-Federico Perno, Francesca Ceccherini-Silberstein, Joseph Fokam.

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
