## [Decision Letter · Decision Letter 0]

7 Jun 2022

PGPH-D-22-00703

HIV-1 Integrase Resistance Associated Mutations and the Use of Dolutegravir in Sub-Saharan Africa: A Systematic Review and Meta-Analysis

Dear Dr. Ngoufack Jagni Semengue,

Thank you for submitting your manuscript to PLOS Global Public Health. After careful consideration, we feel that it has merit but does not fully meet PLOS Global Public Health’s publication criteria as it currently stands. Therefore, we invite you to submit a revised version of the manuscript that addresses the points raised during the review process.

Please submit your revised manuscript by . If you will need more time than this to complete your revisions, please reply to this message or contact the journal office at globalpubhealth@plos.org. Please include the following items when submitting your revised manuscript:

We look forward to receiving your revised manuscript.

Kind regards,

Tsitsi G. Monera-Penduka

Academic Editor

Journal Requirements:

1. Please provide  separate figure files in .tif or .eps format only and remove any figures embedded in your manuscript file.  Please ensure that all files are under our size limit of 20MB.  

For more information about how to convert your figure files please see our guidelines: Once you've converted your files to .tif or .eps, please also make sure that your figures meet our format requirements:

2. Please update the 'Competing Interests' statement in the system with "The authors have declared that no competing interests exist".

3. Please amend your detailed Financial Disclosure statement. This is published with the article. It must therefore be completed in full sentences and contain the exact wording you wish to be published.

a. Please clarify all sources of funding (financial or material support) for your study. List the grants (with grant number) or organizations (with url) that supported your study, including funding received from your institution. 

b. State the initials, alongside each funding source, of each author to receive each grant.

c. State what role the funders took in the study. If the funders had no role in your study, please state: “The funders had no role in study design, data collection and analysis, decision to publish, or preparation of the manuscript.”

d. If any authors received a salary from any of your funders, please state which authors and which funders.

4. Figure 7: please (a) provide a direct link to the base layer of the map used and ensure this is also included in the figure legend; (b) provide a link to the terms of use / license information for the base layer. We cannot publish proprietary or copyrighted maps (e.g. Google Maps, Mapquest) and the terms of use for your map base layer must be compatible with our CC-BY 4.0 license. 

5. In the online submission form, you indicated that "The data that support the findings of this study are available from the corresponding author, upon reasonable request.". All PLOS journals now require all data underlying the findings described in their manuscript to be freely available to other researchers, either 1. In a public repository, 2. Within the manuscript itself, or 3. Uploaded as supplementary information.

6. We have noticed that you have uploaded Supporting Information files, but you have not included a list of legends. Please add a full list of legends for your Supporting Information files after the references list. 

Additional Editor Comments (if provided):

Reviewers' comments:

Reviewer's Responses to Questions

**Comments to the Author**

1. Does this manuscript meet PLOS Global Public Health’s publication criteria? Is the manuscript technically sound, and do the data support the conclusions? The manuscript must describe methodologically and ethically rigorous research with conclusions that are appropriately drawn based on the data presented.

Reviewer #1: Yes

Reviewer #2: Yes

2. Has the statistical analysis been performed appropriately and rigorously?

Reviewer #1: Yes

Reviewer #2: Yes

3. Have the authors made all data underlying the findings in their manuscript fully available (please refer to the Data Availability Statement at the start of the manuscript PDF file)?

Reviewer #1: Yes

Reviewer #2: Yes

4. Is the manuscript presented in an intelligible fashion and written in standard English?

Reviewer #1: Yes

Reviewer #2: Yes

5. Review Comments to the Author

Reviewer #1: The paper is well written, clearly organized and the topic and the findings are important.

my only suggestion is there are a lot of abbreviations and that they do not enhance readability, quite the contrary.

Reviewer #2: Thank you for writing this paper. I have searched for the equivalent a number of times and find myself trying to keep tally on the papers out there. It is a relief and very appreciated that you have done the work and written this review. It's very important. Appreciate the discussion of subtypes and genetic diversity.

To add to the paper, would consider adding some of these points into discussion/intro. They do not need an indepth dive but are all factors that impact this paper and its implications:

1. Could you add more on prevalence of HIV relative to resistance? The total number with HIV and the prevalence are much higher in some regions than others. I think showing this weighting will help those who are working from a policy angle alone. Even just a mention would help.

2. Could you add more on roll out level? This is obviously a dynamic process, but it will be interesting to see how the resistance levels change as roll out is further. Can you also discuss the amount of raltegravir use? How much EVG has been used in Senegal? Clearly other integrase inhibitors are not common choices so interesting if can give a sense of the amount of use to have an idea of the impact.

3. Could you discuss more on HIV-2? I was glad to see this raised. There has always been the question of how EFV based regimens impacted HIV-2 resistance, given the risk of misdiagnosis as HIV-1 in some early testing algorithms and missed dual infection, though not high prevalence still an interesting question.

4. Discussion, any further thoughts on the progression of this. recommendations? model predictions? dynamic process, where see going?

thoughts on models, key observations for what they should include to ensure accurate consideration of baseline and dynamics

https://journals.plos.org/plosmedicine/article?id=10.1371/journal.pmed.1003397

5. anything on dolutegravir and adherence, interruptions, robustness to adherence?

6. Would bring up other meds? ie as given as TLD. this means resistance to TDF is a concern. baseline resistance to other meds? mono therapy risk? any thought not this. without detail, may be worth mentioining

https://pubmed.ncbi.nlm.nih.gov/31895149/

7. Other potential papers

https://journals.lww.com/aidsonline/Fulltext/2021/12152/Adherence,_resistance,_and_viral_suppression_on.4.aspx Adherence, resistance, and viral suppression on dolutegravir in sub-Saharan Africa: implications for the TLD era

Adherence, resistance, and viral suppression on dolutegravir in sub-Saharan Africa: implications for the TLD era Dolutegravir drug-resistance monitoring in Africa (if have access)

https://journals.plos.org/plosmedicine/article?id=10.1371/journal.pmed.1003397

https://pubmed.ncbi.nlm.nih.gov/31895149/

Table 1, Figure 5 and 6. why are first and last names used to note studies?

Figure 5 - could another term be used instead of prevalence for achievement of virology control. It could be misread. frequency, virologic control etc

Figure 7 - would clarify for readers, it makes it look like all of HIV in West Africa is HIV-2. Appreciate HIV-2 and resistance being brought up but would also want to make sure prevalence/distribution clear to readers

6. PLOS authors have the option to publish the peer review history of their article (what does this mean?). If published, this will include your full peer review and any attached files.

**Do you want your identity to be public for this peer review?** For information about this choice, including consent withdrawal, please see our Privacy Policy.

Reviewer #1: No

Reviewer #2: No

---

## [Editor Report · Decision Letter 1]

13 Sep 2022

HIV-1 Integrase Resistance Associated Mutations and the Use of Dolutegravir in Sub-Saharan Africa: A Systematic Review and Meta-Analysis

PGPH-D-22-00703R1

Dear Mr Ngoufack Jagni Semengue,

We are pleased to inform you that your manuscript 'HIV-1 Integrase Resistance Associated Mutations and the Use of Dolutegravir in Sub-Saharan Africa: A Systematic Review and Meta-Analysis' has been provisionally accepted for publication in PLOS Global Public Health.

Best regards,

Kévin Jean

Academic Editor

Dear Authors,

many thanks for having answered every Reviewer's queries.

We have only one last request that we please ask you to address during the production process :

please modify Figures 5 and 6 in order to identify each study by "First author's last name", "Publication year" (don't provide author's first name). [This is the usual way to refer to study in such figures]

Thanks in advance